# Mineralogical controls of the oceanic nickel cycle

Lena Chen [1] ✉, Autum R. Downey[2], Corey Archer[3], Susan H. Little [4], William B. Homoky [1] & Caroline L. Peacock [1]

Transition metals and their isotopes are promising paleo-productivity proxies, but their utility depends on understanding their cycling between sediment and seawater. Using nickel (Ni) as an example, we show how manganese (Mn) minerals control its isotopic composition in oxic marine sediments. By analysing synthetic and natural samples, and simulating sediment diagenesis, we find that most Ni isotope variability in modern Mn-rich sediments is driven by the relative contribution of two bonding mechanisms – adsorption to and structural incorporation into Mn oxides – which evolve during Mn mineral aging and transformation. We also find that isotopically heavy Ni is preferentially released during transformation. This supports a conceptual model where Mn mineral aging and transformation co-modify sediment and seawater Ni isotopes. Using isotope mass-balance we explore the sensitivity of seawater Ni isotope archives to redox change. We suggest that Mn mineral processes are important for any metal isotope proxy whose cycling is coupled to Mn mineral formation.

The current paradigm in marine chemistry emphasises the role of internal oceanic cycling processes (e.g. biological uptake and remineralisation) in determining metal seawater concentrations and isotope distributions[1]. Metal enrichments in the deep ocean compared to the macronutrient phosphate have been explained via reversible scavenging[1–3]. Due to their relatively long oceanic residence time, models of oceanic metal cycling typically neglect oceanic input fluxes[1]. Recent work, however, indicates a potentially important benthic input flux for several metals, resulting from sediment diagenesis[4–6]. In particular, for the trace metal Ni, such a 'recycled' benthic Ni flux may impact deep water Ni concentrations and isotopic compositions and thus play an important but as yet unconstrained role in Ni cycling.

Nickel is bioessential in the oceans. It is a cofactor in multiple enzymes required for metabolic activity, including processes that produce gases such as $CO_2$, $O_2$, and ammonia[7]. As such, Ni is essential to phytoplankton and plays an important role in global carbon, nitrogen, and oxygen cycles, which govern global climate[8]. Nickel also catalyses the production of methane[7]. Therefore, declining

bioavailability of dissolved Ni in the oceans could have contributed to the evolution of Earth's atmosphere, from methane-dominant in the Archaean to more oxygen-rich in the Proterozoic[9]. Given its biological importance on both modern and ancient Earth, there is active research developing Ni and its isotopes as tracers for key biogeochemical processes in the marine environment[10]. Its utility as a biogeochemical tracer, however, is predicated on a robust understanding of the balance between Ni sources and sinks in the ocean, where critical knowledge gaps remain.

While the dominant external source of Ni to seawater is input from rivers[11,12], the dominant sink of Ni from seawater is removal to dispersed Fe–Mn (oxyhydr)oxide phases in oxic sediments[4,5,13]. Assuming steady state, Ni input from rivers and output to oxic sediments should balance, along with their flux-weighted isotopic compositions[4,5,14]. The Ni riverine flux and its isotopic composition is relatively well-constrained, at 0.37 Gmol/Yr and 0.8‰[11,12] ($\delta^{60}$Ni, the parts per thousand deviation of the $^{60}$Ni/$^{58}$Ni ratio from NIST SRM 986), while estimates of the Ni output to oxic sediments based on the Ni/Mn ratio of Fe–Mn

[1]School of Earth and Environment, University of Leeds, Leeds LS2 9JT, UK. [2]Department of Earth and Space Science, University of Washington, 4000 15th Ave NE, Seattle, WA 98195, USA. [3]Institute of Geochemistry and Petrology, Department of Earth Sciences, ETH Zürich, Clausiusstrasse 25, 8092, Zürich, Switzerland. [4]Department of Earth Sciences, University College London, Gower Street, London WC1E 6BT, UK. ✉e-mail: lena.chen@bristol.ac.uk

crusts and pelagic sediments are 2–3 times larger than the riverine flux[4,5]. To balance this budget, a significant recycled benthic flux of Ni from sediments has been proposed[4,5,15,16]. Alternatively, hydrothermal Mn-rich sediments with a lower Ni/Mn ratio than Fe-Mn crusts and pelagic sediments may contribute to the overall sediment sink[13]. In addition to the flux imbalance, the $\delta^{60}$Ni values of oxic sediments range from −1.49 to +2.47‰[4,5,13,17–19]. The flux imbalance and large range of $\delta^{60}$Ni for Mn-rich sediments indicate that there are unknown, fundamental processes occurring in oxic marine sediments controlling Ni and Ni isotope ratios that need further examination[4,13,20].

Mn-rich oxic sediments are made up of a mixture of poorly but variably crystalline layered-structure phases (phyllomanganates), from most poorly crystalline vernadite (often interchangeably referred to as $\delta$MnO$_2$, a synthetic analogue) to c-disordered birnessite[21,22]. These minerals can take up Ni via two distinct bonding mechanisms, namely, surface adsorption and structural incorporation. Experimental studies with variably crystalline phyllomanganate show that Ni is adsorbed to the mineral surface, above/below vacancy sites in the mineral layers[21,23,24], and at seawater pH and with time is incorporated into the mineral structure[25]. A computational study using constructed phyllomanganate nanosheets suggests that Ni can be adsorbed to the mineral surface, at layer edge sites, and incorporated into the structure during mineral growth[26]. As such, the fraction of Ni that is adsorbed versus incorporated is expected to change during diagenetic aging[25]. This evolution is important because, if Mn-rich oxic sediments are at isotopic equilibrium, the Ni bonding mechanism that provides the strongest bonding environment (shortest, stiffest bonds), where the energy state is minimised, should concentrate the heavier isotope[27]. Previous work with experimental samples and natural ferromanganese deposits shows that incorporated Ni is exchangeable with solution[25,28] and that incorporated Ni occupies a lower energy state than adsorbed Ni[23]. Thus, incorporation should increase the $\delta^{60}$Ni value of Mn-rich sediments if there is isotopic exchange with an aqueous phase. Accordingly, experiments find that adsorbed Ni is enriched in lighter isotopes, with a Ni isotope fractionation ranging from $\Delta^{60}$Ni$_{mineral-aqueous}$ = −1.15 to −4.02‰[24,29] ($\Delta^{60}$Ni$_{mineral-aqueous}$ = $\delta^{60}$Ni$_{mineral}$ − $\delta^{60}$Ni$_{aqueous}$). While in Fe–Mn crusts, where Ni is incorporated[30], $\delta^{60}$Ni is high, at about +1.5‰[17–19], compared to seawater, at +1.3‰[31,32]. Furthermore, $\delta^{60}$Ni values of Mn-rich deposits correlate with Co/Mn ratios, such that their isotopic composition may in part be controlled by accumulation rate[13]. At slower accumulation rates, Ni is increasingly incorporated, and Ni isotope compositions are consequently heavier[13]. Taken together, these observations suggest that the balance between Ni adsorbed versus incorporated in Mn minerals plays a critical role in determining the Ni isotope composition of oxic sediments[13]. The relationship between Ni bonding mechanisms and Ni isotope compositions, however, has never been systematically tested.

Post deposition, phyllomanganates undergo diagenetic transformation into tunnel-structure Mn minerals (tectomanganates), typically through oxic/suboxic diagenesis and/or under mild hydrothermal conditions[15,33–35]. As such, todorokite is often the dominant phase in hydrothermal Mn nodules[18] and is also observed in sub-oxic diagenetic settings[35]. This transformation may therefore further modify the proportion of Ni that is adsorbed versus incorporated and influence the amount and isotopic composition of Ni that is either retained in sediments or recycled back into the ocean[4,15]. Simulated transformation experiments show that transformation from phyllomanganate to tectomanganate todorokite releases a substantial fraction of Ni into solution[15]. Buried todorokite-bearing nodules also contain less, and isotopically lighter, Ni than surface phyllomanganate-bearing nodules[4,35]. It is therefore likely that post-depositional diagenetic transformations can exert an additional control on the Ni isotope budget of Mn-rich sediments that is critically uncharacterised.

Here, we show the importance of bonding mechanisms and sedimentary diagenetic processes that alter bonding mechanisms in controlling Ni isotope ratios in experimentally synthesised Ni-birnessite and natural Mn-rich sediment samples. Using a wet chemical treatment, we operationally determine the fraction of adsorbed versus incorporated Ni in our synthetic and natural samples and evaluate the relationship between bonding mechanisms and Ni isotope compositions. We further transform experimental Ni-phyllomanganate into todorokite via a mild reflux procedure designed to mimic natural Mn oxide transformations at the seafloor[15,33] and report its effects on Ni concentrations and Ni isotope ratios over time. Consequently, we present a conceptual model for Ni isotope ratio evolution in oxic sediments, which unifies our experimental constraints with observations from natural samples. Our model explains the vast range in isotopic compositions measured in Mn-rich sediments and suggests an alternative mechanism for deepwater enrichments of trace metals in the ocean.

## Results and discussion
### Bonding mechanisms to Mn minerals explain sedimentary Ni isotope compositions

To examine the importance of bonding mechanisms controlling Ni isotope ratios, we prepare synthetic Ni-birnessite samples at different precipitation rates to encourage different fractions of adsorbed versus incorporated Ni, where at a slower precipitation rate, there is a greater fraction of incorporated Ni. We then use a wet chemical treatment to operationally define the fraction of adsorbed versus incorporated Ni in our synthetic Ni-birnessite samples (see Methods). The treatment is also applied to eight Mn-rich sediments with known Ni isotope compositions[4,13] (Supplementary Information 1). The eight natural samples include six open ocean deep-sea pelagic sediments from the Pacific Ocean, representing hemipelagic sedimentation, and two samples from the MANOP sites located in the Guatemala Basin, representing hemipelagic (Site H) and hydrothermal (Site M) sedimentation, respectively. The other two natural samples are the USGS Mn nodule reference materials NOD P1 and NOD A1.

To validate the wet chemical treatment, we subject three synthetic Ni-birnessite samples to extended X-ray absorption fine structure (EXAFS) spectroscopy (see Methods). Our treatment indicates adsorbed fractions of $0.70 \pm 0.06$, $0.52 \pm 0.04$ and $0.39 \pm 0.06$ Ni, which agree very well with EXAFS spectroscopy, determining adsorbed fractions of $0.73 \pm 0.04$, $0.60 \pm 0.05$ and $0.35 \pm 0.04$ Ni, respectively (Supplementary Information 2). We thus have confidence that our treatment provides a good estimate for adsorbed versus incorporated Ni in synthetic phyllomanganates. We also subject NOD P1 and NOD A1 to EXAFS spectroscopy, where our treatment indicates adsorbed fractions of $0.36 \pm 0.02$ and $0.22 \pm 0.02$ Ni, respectively. These results agree reasonably well with those from spectroscopy, determining that a small amount of Ni is adsorbed for NOD P1 ($0.20 \pm 0.05$) and a slightly smaller amount for NOD A1 ($0.10 \pm 0.04$) (Supplementary Information 2). Comparing our wet chemical treatment to spectroscopy suggests a discrepancy in the adsorbed fraction of up to -0.2 for natural sediments. This discrepancy may be because our treatment removes adsorbed Ni that cannot be resolved with spectroscopy, for example, Ni loosely bound via electrostatic interactions and/or bound with mineral-associated organic matter[36]. Or it may indicate that our treatment overestimates the adsorbed fraction by removing adsorbed but also a minor amount of incorporated Ni. Given the heterogeneity of natural sediments, it is also possible that for other samples our treatment might underestimate the adsorbed fraction, by failing to completely remove Ni strongly bound during surface complexation[36]. A minor amount of Ni may also be associated with other phases in the Mn nodules and may be removed during the treatment. To represent the discrepancy between our treatment and spectroscopy, we consider the adsorbed fraction of our natural sediment samples as

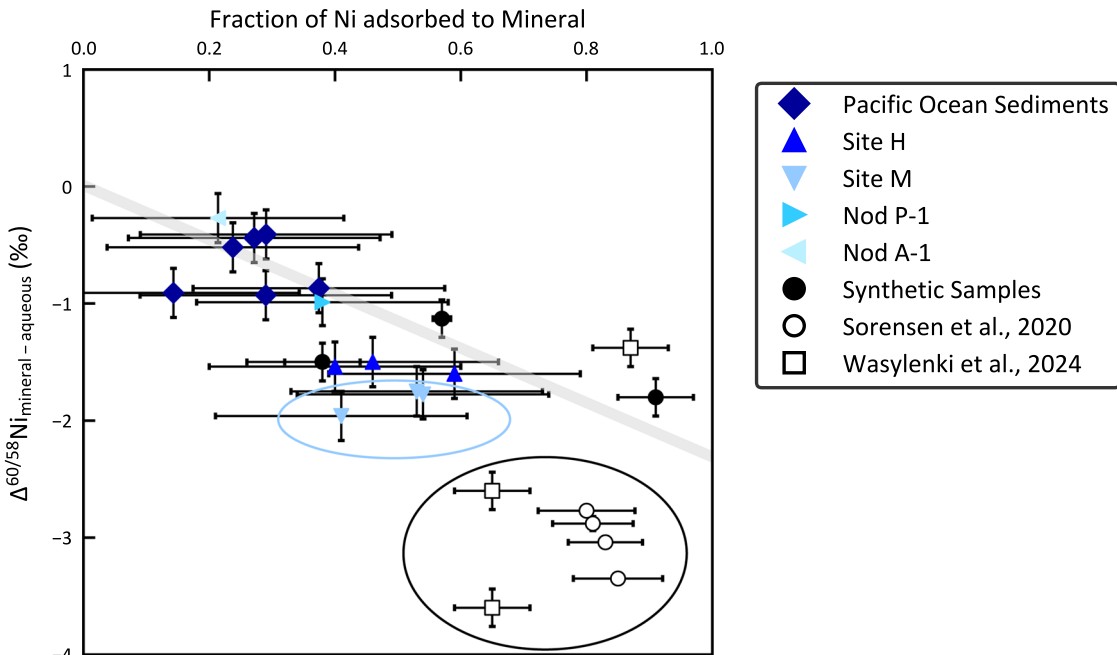

**Fig. 1 | Fraction of Ni adsorbed vs $\Delta^{60}Ni_{mineral-aqueous}$ (‰) for synthetic samples and natural sediments from this study, and synthetic samples from the literature (Sorensen et al.[24] and Wasylenki et al.[29]).** Fraction of Ni adsorbed was determined for our synthetic samples and natural sediments by this study using wet chemical treatment, and for the synthetic samples from the literature by those studies using EXAFS spectroscopy. Isotope ratios were determined for our synthetic samples by this study, and for natural sediments (Pacific Ocean sediments, MANOP Site H and M, and USGS NOD A1 and NOD P1) by previous work[4,13]. Isotopic fractionation for natural sediments is calculated relative to the mean value of seawater at 1.33 ± 0.13‰[32]. Isotopic ratio data for the synthetic samples from the literature were determined by those studies. x-Axis error bars for our synthetic samples represent the standard deviation from three repeated wet chemical treatments, where the fraction of Ni that is adsorbed agrees well between our treatment and EXAFS spectroscopy. x-Axis error bars for natural sediments represent a ±0.2 discrepancy between the fraction of Ni adsorbed as determined by wet chemical treatment vs that determined by EXAFS spectroscopy of the USGS nodules. y-Axis error bars are the long-term reproducibility of the isotope ratio analyses (±0.08‰) combined with the variation of isotope ratios in seawater (±0.13‰)[31]. The grey bar represents a York linear regression fit to our equilibrium-dominated synthetic samples and natural sediments ($r = 0.64$, $p < 0.01$). Circled data in black highlights the kinetic-dominated synthetic samples from the literature, and in blue highlights the MANOP Site M samples.

determined by our treatment to be ±0.2, which accounts for a conservative scenario in which the treatment might overestimate or underestimate the adsorbed fraction.

Considering our synthetic samples, natural sediments and available synthetic data from the literature[24,29], these appear to express a combination of equilibrium and kinetic isotope effects. Determining the relative contributions of these two effects is difficult but there appear to be two sets of data—the first encompasses our synthetic samples, natural sediments and one of the literature data points (plotting around $\Delta^{60/58}Ni_{mineral-aqueous}$ = -0 to −2‰, Fig. 1). We suggest that these data reflect both equilibrium and kinetic isotope effects but might be nearer to equilibrium isotope fractionation. A second group of very negative $\Delta^{60/58}Ni_{mineral-aqueous}$ values from the literature (plotting around $\Delta^{60/58}Ni_{mineral-aqueous}$ = ~−3‰, circled in black, Fig. 1) seem to be further from equilibrium and mostly governed by kinetic isotope fractionation.

To distinguish between equilibrium and kinetic isotope fractionation in simple sorption experiments, experimental data are either fit to two parallel lines or a Rayleigh fractionation curve, respectively. Considering the literature experimental data first, in the study of Sorensen et al.[24], where all the data plot in the second group of very negative $\Delta^{60/58}Ni_{mineral-aqueous}$ values, $Ni_{(aq)}$ is mixed with pre-formed phyllomanganate with a relatively short contact time (24 h). Nickel is mostly adsorbed onto the mineral surface, with a large negative isotope fractionation (~−3‰, circled in black, Fig. 1)[24]. Due to the small range in % Ni adsorbed in their experiments, the data is not fitted to either model. From previous work with other metals, however, concentration equilibrium is achieved more quickly than isotopic equilibrium[37]. Thus, although isotopic exchange between Ni in solution

and Ni at the mineral surface might be relatively fast, the contact time in these experiments may not allow isotopic equilibrium to be reached, and therefore it is likely that these data express a kinetic isotope effect[29]. In the study of Wasylenki et al.[29], $Ni_{(aq)}$ is again mixed with pre-formed phyllomanganate. The two data points generated at high ionic strength plot in the second group of very negative $\Delta^{60/58}Ni_{mineral-aqueous}$ values (~−3‰, circled in black, Fig. 1), while the data point generated at low ionic strength plots in the first group of data points (plotting around $\Delta^{60/58}Ni_{mineral-aqueous}$ = -0 to −2‰, Fig. 1). At high ionic strength more Ni is incorporated into the structure, and the isotopic fractionation evolves from a large negative value at 49 h (~−3.6‰, circled in black, Fig. 1) to a smaller negative value at 27 days (~−2.6‰, circled in black, Fig. 1)[29]. At low ionic strength after 67 h, Ni is mostly adsorbed onto the mineral surface, with a moderate negative isotope fractionation (~−1.4‰, Fig. 1). Incorporation is expected to induce a kinetic isotope fractionation due to slow dehydration and diffusion of surface Ni into the mineral structure and the kinetic effect should therefore subside with time as isotopic exchange gradually occurs between the surficial and structural Ni[29]. Thus, the high ionic strength data (~−3.6‰ and ~−2.6‰, circled in black, Fig. 1) are also interpreted as expressing a mostly kinetic isotope effect, and there is a better fit to Rayleigh fractionation curves. While for the low ionic strength data point (~ −1.4‰, Fig. 1), the kinetic isotope effect is likely smaller, as the data show a closer alignment with parallel lines[29].

In nature, a large kinetic isotope effect should therefore be expressed in natural Mn-rich samples where Ni uptake is fast and/or results in Ni incorporation, without sufficient time to reach isotopic equilibrium. Indeed, this effect appears to be observed in the Black Sea water column. Here, intense Mn-Ni cycling occurs, such that Mn

oxides formed in the oxic layer sink below the redoxcline, are reductively dissolved, and dissolved Mn rises through the water column to be rapidly reprecipitated[38]. As such, it is likely that Mn particles are not only coprecipitated with Ni as they are rapidly reprecipitated but that they also continue to adsorb Ni as they sink towards the redoxcline. Accordingly, variations in dissolved $\delta^{60}$Ni values across the Black Sea redoxcline might reflect a mostly kinetic isotope effect of $\Delta^{60/58}$Ni$_{mineral-aqueous} = -4‰$[39].

It is expected that most natural marine sediments are at or nearer to isotopic equilibrium, because there is sufficient time for isotopic exchange between Ni in solution and Ni at the mineral surface, and between Ni at the surface and Ni in the mineral structure[29]. The similarity between the natural sediments and our synthetic samples, and the distinction of both from the literature data, suggests that most of our synthetic samples show a much smaller contribution from kinetic isotope effects. We suggest this because, unlike the Sorensen et al.[24] and Wasylenki et al.[29] studies, our synthetic samples are prepared via coprecipitation (rather than by mixing Ni$_{(aq)}$ with pre-formed phyllomanganate). Although we cannot fit our data to two parallel lines or a Rayleigh fractionation curve, coprecipitation may accelerate the transfer of Ni from the mineral surface into the structure, thus facilitating isotopic exchange and a more rapid approach to isotopic equilibrium. Furthermore, if Ni enters the structure as the mineral is forming via adsorption to the layer edge sites, this should bypass the slow dehydration and diffusion of surface Ni into the structure and thus limit the kinetic isotope effect in our experimental data[26]. The strength of the fit through our data relies upon the combination of the natural and our synthetic samples; however, one of our data points falls notably below the trend line. We think this is because, two samples that are mostly adsorbed show little kinetic isotope effect (e.g. our synthetic sample with the least Ni incorporated (~90% Ni adsorbed) has a similar $\Delta^{60}$Ni$_{mineral-aqueous}$ to the low ionic strength literature sample in which the kinetic isotope effect is likely small (~$-1.8‰$, compared to ~$-1.4‰$, Fig. 1)), whereas one sample that is mostly incorporated shows one of two contributing effects: either a kinetic isotope effect that lowers $\Delta^{60/58}$Ni$_{mineral-aqueous}$ because Ni is adsorbed to vacancy sites and incorporated and as such there is a time lag associated with diffusion; or if the Ni is adsorbed to layer edge sites and becomes incorporated as the mineral grows[26], while there would be minimal time lag, this could express the negative $\Delta^{60/58}$Ni$_{mineral-aqueous}$ that occurs during edge-sharing adsorption[40]. The deviation of this sample from the trend line is within the range of the deviations that we observe in the natural samples. As such, it is consistent with the overall interpretation of our results.

Despite the difficulties in replicating Ni uptake in natural sediments using necessarily much shorter-term synthetic samples, we find that the range of isotopic compositions measured in the synthetic samples and natural Mn-rich sediments can be explained by a mixing relationship between the two different bonding mechanisms, where Ni is either adsorbed or incorporated (Fig. 1). Crucially, this mixing relationship provides a mechanistic explanation for the vast range of Ni isotope compositions measured in natural sediments. For example, if Ni in Fe–Mn crusts is essentially entirely incorporated[30], our model predicts a fractionation of approximately +0.0‰ relative to seawater, and although there is a large range of crust isotopic compositions from 0.76 to 2.47‰, two thirds of crust data fall between 1.2 and 1.8‰ with an average $\delta^{60}$Ni of about +1.5‰, which is isotopically fractionated relative to seawater by about +0.2‰[17,19]. This suggests that for most Fe–Mn crusts, the bonding mechanism of Ni is similar, while the isotope values at the extremes of this range might reflect other bonding mechanisms to phyllomanganates or bonding to other mineral phases, as discussed below for our natural sediments. Minor amounts of Ni can be incorporated in the structure of Fe (oxyhydr)oxides in Fe–Mn crusts[41], particularly when the Mn layers are intermixed with Fe (oxyhydr)oxides. Incorporation of Ni into Fe (oxyhydr)oxides[41] is

suggested to contribute to the particularly heavy isotope compositions observed in some Fe–Mn crusts[17]. Also, Ni isotope values that fall outside the range may be influenced by alteration processes occurring in the crust[19].

Natural marine sediment Ni isotope compositions that deviate from the mixing line could reflect a range of secondary factors. In the case of natural marine phyllomanganates that are mineralogically alike to our synthetic samples, experimental studies show that Ni adsorption occurs mainly at vacancy sites[21,23,24], while a computational study suggests that Ni adsorption also occurs at layer edge sites[26]. For phyllomanganates that are mineralogically different to our synthetic samples, like those without vacancies[21] and tectomanganates formed during early diagenesis[15], adsorption to layer edge sites is likely the dominant adsorption mechanism. Within the -0.2 uncertainty, our wet chemical treatment should remove adsorbed Ni, regardless of bonding environment (i.e. vacancy or layer-edge). Because bonding environment influences isotopic fractionation[27,41,42], Ni bonded via different adsorption mechanisms may exhibit differing isotopic fractionation and hence lead to deviations from the mixing line[43]. Similarly, Ni may be adsorbed by other mineral phases such as Fe (oxyhydr)oxides and clays, which also present different bonding environments and thus Ni isotope fractionation during adsorption[42,44]. Adsorption of Ni to Fe (oxyhydr)oxides and clays[41,42] is suggested to contribute to the particularly light isotopic compositions observed at MANOP site M, where all samples from the site show a negative deviation from our mixing line (circled in blue, Fig. 1)[4].

## Transformation of Mn minerals modifies bonding mechanisms and thus sedimentary Ni isotope compositions

In freshly precipitated phyllomanganate phases, Ni is adsorbed at the surface and progressively incorporated into the structure[25], minimising the energy state[23]. As such, Ni should become increasingly isotopically heavy over time if in exchange with an aqueous phase. Following this trend, oxic marine sediments should eventually acquire the same isotopic composition as Fe–Mn crusts, as isotopic exchange occurs with porewaters. The Ni isotope compositions in Mn-rich pelagic sediments measured thus far, however, remain isotopically light compared to Fe–Mn crusts, and relatively constant downcore[13]. Although the prevalence of todorokite is difficult to discern in dispersed Mn-oxide marine sediments[4], one explanation for this observation is that Mn mineral transformation could be occurring in these sediments, interrupting the predicted bonding evolution from adsorbed to incorporated Ni and thus arresting the isotopic evolution of the phyllomanganate phase. To investigate this hypothesis, we transformed Ni-birnessite into todorokite via a mild reflux procedure designed to mimic early sediment diagenesis and low-temperature hydrothermal conditions[15,33]. During the reflux procedure, aliquots of the solid-solution suspension were taken at selected time points, then purified and measured for their Ni isotope ratios (see Methods). A full description of the birnessite to todorokite transformation is provided in Supplementary Information 3.

During transformation, isotopically heavy Ni is released to solution from the solid phase (Fig. 2). We observe an initial release of isotopically heavy Ni during transformation, heavier by $1.87 \pm 0.16‰$ than the solid phase (Fig. 2 A, B), after which the isotopic difference between solution and mineral decreases sharply to $0.30 \pm 0.16‰$ (Fig. 2A, B). Then, after 14 days, while there is no further release of Ni to solution (Fig. 2C, D), $\Delta^{60}$Ni$_{aqueous-mineral}$ increases gradually from $0.59 \pm 0.16‰$ to $0.94 \pm 0.16‰$ at 28 days (Fig. 2A, B). The initial release of isotopically heavy Ni coincides with the onset of todorokite nucleation at ~6 h observed using transmission electron microscopy in our previous work[15], suggesting that isotopically heavy incorporated Ni from the phyllomanganate is released to facilitate kinking of the phyllomanganate layers, necessary to initiate todorokite formation. The following sharp decrease in $\Delta^{60}$Ni$_{aqueous-mineral}$ is coincidental with

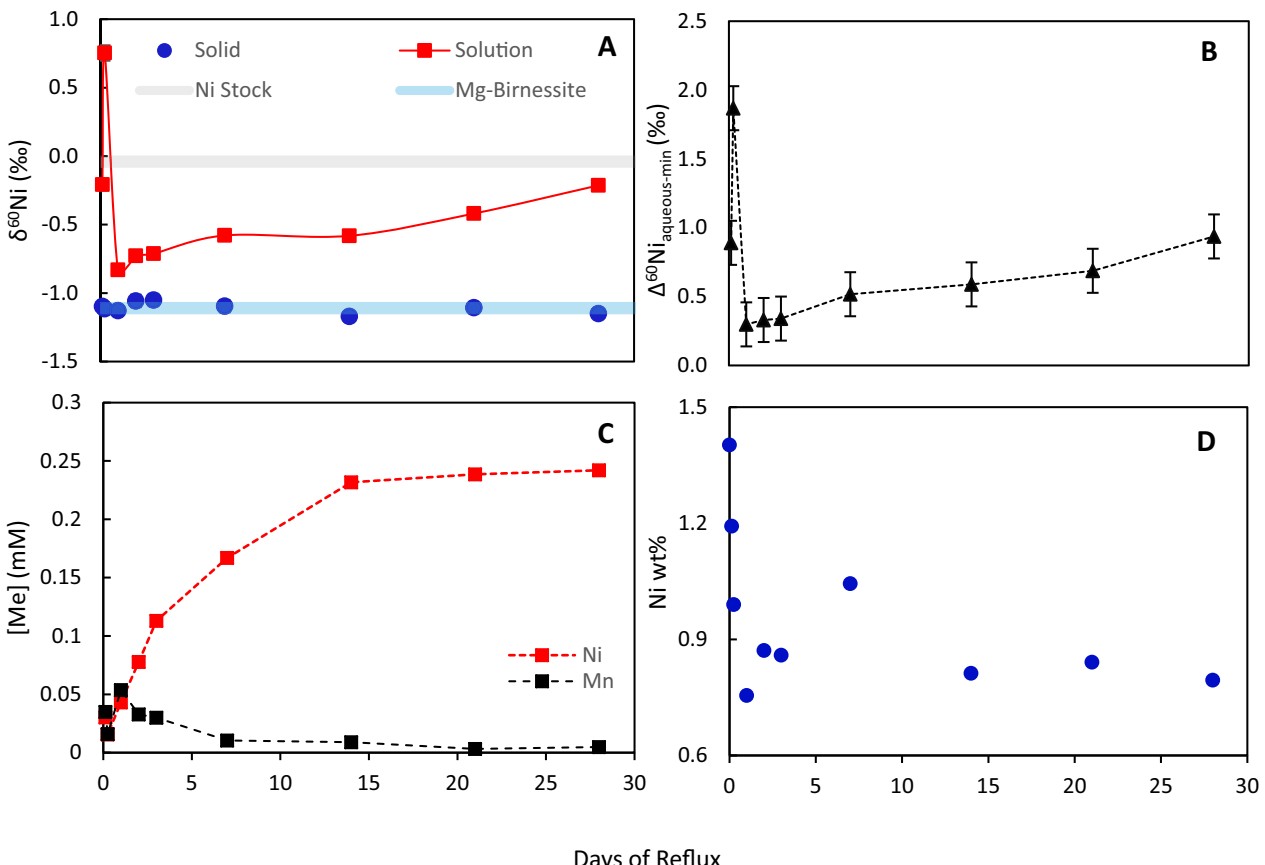

**Fig. 2 | Ni concentration and isotope measurements of the solid and solution products from the transformation reflux procedure.** $\delta^{60}Ni$ of the solid and solution products from time aliquots of the transformation reflux procedure (**A**) and the $\Delta^{60}Ni_{aqueous-min}$ as a function of time (**B**). Concentration of Ni and Mn in the reflux solution as a function of reflux time (**C**) and the absolute Ni wt% of the solid as a function of reflux time (**D**). Error bars on $\delta^{60}Ni$ values are the long-term reproducibility of the isotope ratio analyses at ±0.08‰ and are within the size of the symbols. Error bars on Mn and Ni concentrations were less than ±1.32% and are within the size of the symbols.

a spike in Mn concentration in solution (Fig. 2C). These features coincide with the onset of some phyllomanganate dissolution observed in our previous work[15], suggesting that isotopically light Ni associated with the phyllomanganate is released during the dissolution step. The final increase in $\Delta^{60}Ni_{aqueous-mineral}$ after 14 days and the lack of further release of Ni into solution coincides with the crystal ripening step observed in our previous work[15], during which time any Ni released during dissolution of smaller crystals is likely re-adsorbed onto larger crystals forming. Therefore, the increase in solution $\delta^{60}Ni$ could be due to a kinetic isotope effect during re-adsorption of isotopically light Ni. Our previous work by EXAFS spectroscopy shows that Ni uptake to the neo-todorokite is via adsorption as inner-sphere complexes located at the edge sites of the todorokite walls. These complexes are octahedral, with no change in the Ni coordination number[15], such that the $\delta^{60}Ni$ of the solution should stabilise once equilibrium is reached. Overall, at the end of the experiment, we observe an isotopic fractionation between the solution and mineral of $\Delta^{60}Ni_{aqueous-mineral} = +0.94 \pm 0.16$‰.

**Mn mineralogical processes govern the Ni isotope composition of marine sediments**

The data presented in Fig. 1 suggest that the Ni isotope composition of marine sediments can be explained by the proportion of isotopically light adsorbed Ni versus isotopically heavy incorporated Ni in Mn-rich phyllomanganate minerals. Figure 2 indicates that, as Ni is redistributed during the transformation of these minerals, isotopically heavy Ni is lost to solution. Therefore, Mn mineral

transformation appears to interrupt the progressive incorporation of isotopically heavy Ni into phyllomanganate vacancies and might explain why Ni isotope compositions in Mn-rich pelagic sediments remain isotopically light, and approximately constant downcore[13]. We suggest that as Mn-rich precipitates are buried in the sediment column and undergo diagenesis, their Ni bonding mechanisms and thus their Ni isotope compositions are altered. We use diagenesis here collectively to refer to the processes shown graphically in Fig. 3. Process 1 aims to represent 'aging' of the phyllomanganate phase, during which Ni is first adsorbed at the surface and then incorporated[25], becoming increasingly isotopically heavy over time if there is isotopic exchange with seawater (Fig. 3, process(1)). Process 2 aims to represent 'transformation' of the phyllomanganate phase, where, while Fe–Mn crusts may reach the end-point of the ageing process (i.e., 100% incorporated Ni), Mn-rich sediments are subject to mineral transformation post-deposition (Fig. 3, process (2)). Transformation to todorokite releases isotopically heavy incorporated Ni to porewaters, which interrupts the isotopic evolution of the precipitates. We note it might be the onset of this transformation, where incorporated Ni is ejected from the mineral, that is most crucial in altering Ni isotope compositions in marine sediments. To what extent complete transformation occurs in marine sediments is uncertain, as the presence of todorokite in marine sediments is often difficult to establish[4]. Process 3 then aims to represent 'remobilisation' of the released Ni. The released Ni may be re-adsorbed to the neo-formed todorokite, likely as an outer-sphere complex or octahedral inner-sphere complex, where an outer-sphere complex is

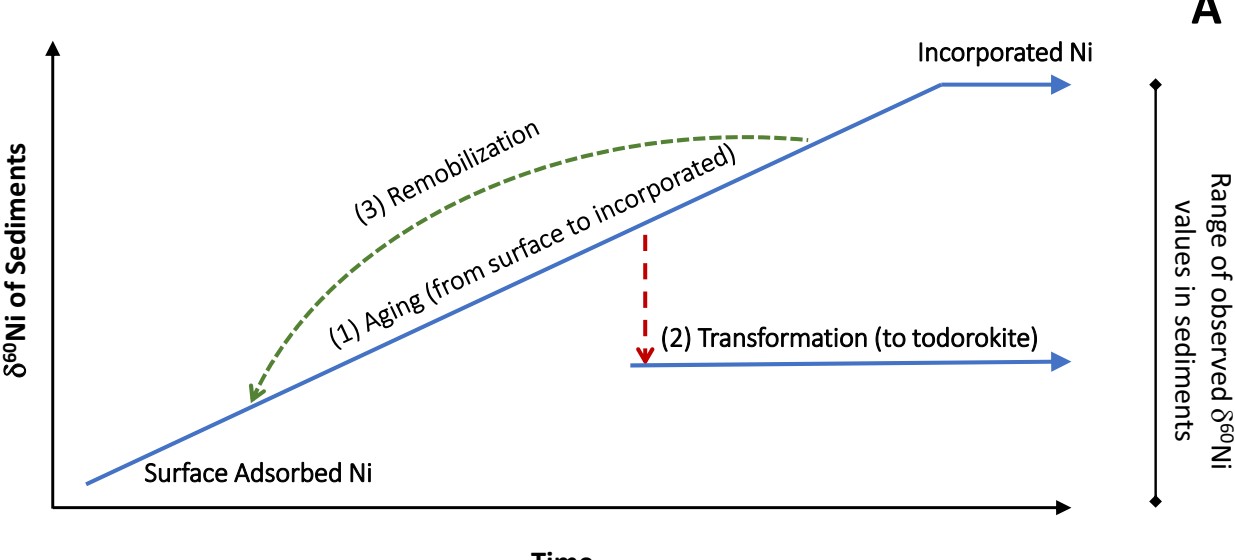

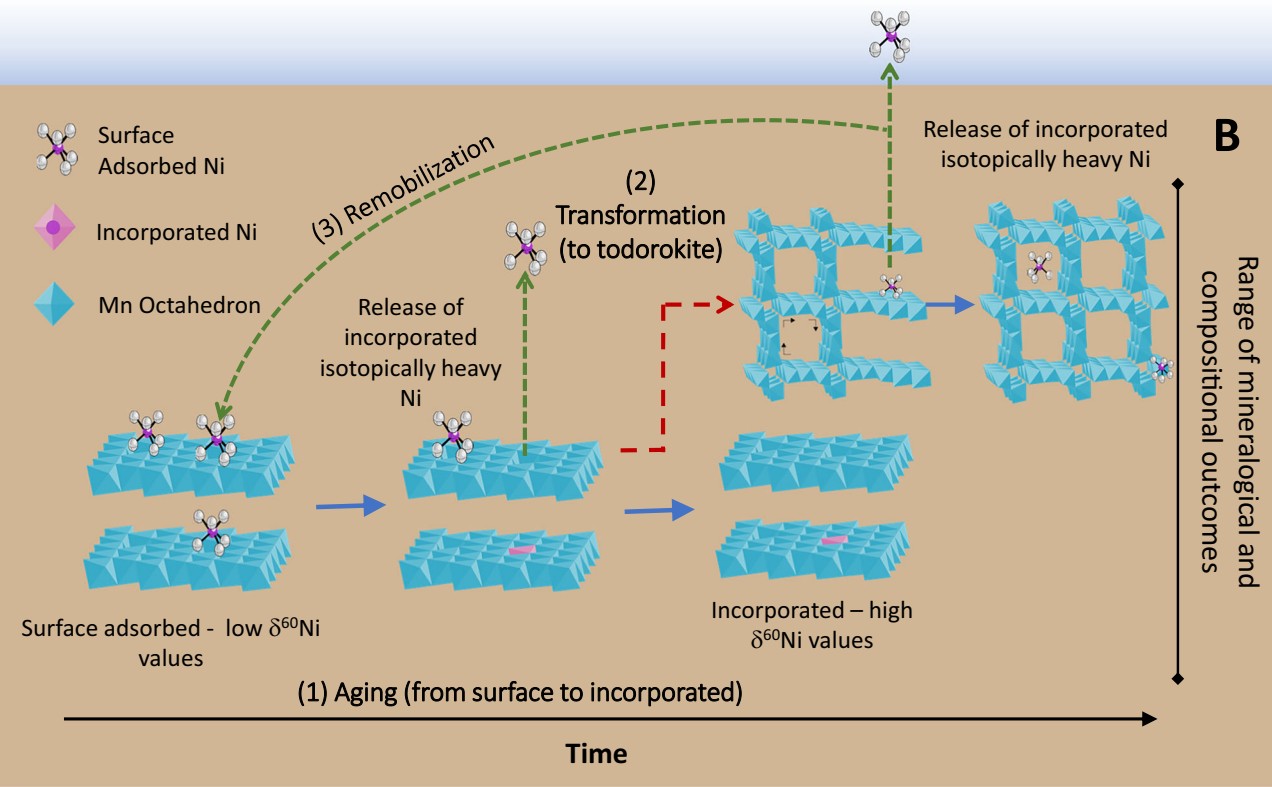

**Fig. 3 | Conceptual model of the Ni isotope and mineralogical evolution of birnessite in marine sediments.** Conceptual model of the Ni isotope (**A**) and mineralogical (**B**) evolution of birnessite in marine sediments, whereby Mn mineral ageing and transformations co-modify the Ni isotope compositions of sediments and seawater. Three processes are outlined: (1) Ageing of birnessite, during which adsorbed Ni becomes incorporated; (2) Transformation to todorokite, during which incorporated Ni is released; (3) Remobilisation of Ni, in which Ni is released during transformation to todorokite or during reductive dissolution, and may then be re-adsorbed or released to porewaters.

expected to induce minimal isotope fractionation [37,45,46]. The released Ni may also be re-adsorbed to the authigenic birnessite as a surface adsorbed complex, which induces a negative isotope fractionation, and/or be released into porewaters. The transformation and re-adsorption process 'resets' the isotopic composition of the precipitates, effectively counterbalancing the trend towards heavier isotopic compositions over time and maintaining an approximately constant and relatively light isotopic composition downcore.

In sediments where Mn oxides are reduced and Mn is released to the porewaters [47], mineral dissolution may also remobilise Ni, which is re-adsorbed onto freshly precipitated Mn oxides in the shallower oxygenated zone. This release and reset process thus explains the lower amounts of incorporated Ni and lighter isotopic compositions of suboxic/reducing compared to pelagic/oxic sediments [4]. Overall, we posit that the range of $\delta^{60}Ni$ values observed in Mn-rich marine sediments is a function of the bonding mechanism, which is controlled by

the extent to which the minerals are subject to early diagenetic mineralogical processes (Fig. 3).

## Sedimentary controls on seawater Ni distributions and Ni isotope compositions

The burial of Ni in marine sediments exerts a primary control over the oceanic concentration and isotopic composition of Ni. We find that the transformation of Mn minerals during sediment diagenesis releases isotopically heavy Ni to porewaters, which, under favourable conditions (i.e. via diffusion, advection, or bioturbation[48]), could be returned to bottom waters. While detecting the impact of the recycled Ni flux on water column Ni isotope compositions is likely to be challenging against high background deep water Ni concentrations, an isotopically heavy, recycled benthic flux has been invoked to account for the mass imbalance in the global oceanic Ni budget[4,5], and is supported by porewater and solid phase data[6]. Furthermore, recent work has highlighted the enrichment of Ni relative to the macronutrients phosphate and silicate in the modern deep ocean, and attributed this enrichment to reversible scavenging (i.e. the uptake of Ni to particles in the water column and then the release of some fraction of this Ni from these particles as they sink)[1]. Here, we suggest that Ni released during diagenesis may also contribute to the deepwater enrichment of Ni. Future models of the marine Ni cycle should consider the role of sediment diagenesis in determining its oceanic distribution[49,50].

Our understanding of the oceanic Ni sink requires a re-evaluation of the use of Ni isotope ratios recorded in marine sediment archives as a paleoenvironmental proxy. Nickel has an oceanic residence time of ~20,000 years, which exceeds the timescale of the global ocean overturning circulation (~1000 yrs)[4,13], so that Ni isotope compositions recorded in open marine sediment archives should represent a well-mixed oceanic Ni reservoir. Variations in Ni removal mechanisms from the ocean reservoir to the sediments over time, however, impart unique isotope effects, which are reflected in the seawater Ni isotope composition and consequently the sediment record. Throughout Earth's history, changes in ocean redox will have caused changes in the global distribution of sediment redox conditions, which in turn will have altered the balance of oceanic sinks for Ni[51]. As such, two comparatively small components of the oceanic sink today, euxinic and reducing sediments[14,39], may have had a greater contribution during periods of expanded ocean deoxygenation[51]. Furthermore, we have shown that the Ni isotope composition of the oxic sediment sink responds to diagenesis, a process that might have varied in the past; for example, periods of increased Mn mineral transformation might be linked to tectonic drivers of seafloor hydrothermalism[52,53]. The utility of Ni isotopes as a paleoenvironmental proxy, therefore, requires consideration of their sensitivity to these variations through time.

If we assume that the isotopic input of Ni to the ocean from weathering (0.8‰) has remained constant, we can use the modern relationships between Ni in sediments and seawater to illustrate the sensitivity of seawater $\delta^{60}Ni$ to theoretical perturbations of oceanic Ni removal (Fig. 4). Using this approach, we find seawater $\delta^{60}Ni$ values are sensitive to ocean redox changes, and more sensitive to oxic-reducing variations than to oxic-euxinic variations (Fig. 4A). This finding is consistent with the fact that Ni is almost unreactive to sulphides[39] and therefore relatively insensitive to increasing euxinia. Isotopic compositions of other metals, notably Mo and U, that are used in tracing global redox evolution[51], are more sensitive indicators of marine euxinia. Therefore, by virtue of being a non-chalcophile element, Ni isotopes might complement these more established redox proxies, where isotope excursions to lighter values might indicate more reducing conditions without the necessary onset of euxinia.

Future applications of Ni isotopes as a redox proxy however, also require consideration of the variable isotopic fractionation between Mn oxides and seawater (Fig. 4B). Using the range of $\Delta^{60}Ni_{oxsed-sw}$ for Mn-rich sediments as a function of bonding mechanisms (Fig. 1), we

show how seawater $\delta^{60}Ni$ would change assuming modern proportions of Ni removal to the respective sinks (Fig. 4B). We note that, on average, the endmember scenarios of either complete incorporation or adsorption are unlikely. Higher temperatures increase the rate of mineral transformation[52], however, therefore periods of more extensive seafloor hydrothermalism could have theoretically decreased $\Delta^{60}Ni_{oxsed-sw}$ and generated an isotopically heavier ocean.

Global oceanic cycling of transition metals is governed by the balance between sources and sinks, and long-term changes to either will change their mean oceanic concentration and isotopic composition. Evidently, for Ni, Mn mineral ageing and transformations can exert an important control on the Ni isotope compositions of sediments and seawater, which leads us to caution against the interpretation of upper ocean biogeochemical controls of Ni isotope records without considering the oxic sediment sink and marine redox conditions. More broadly, several other bio-essential metals, including Co, Zn and Cu, are strongly associated with and potentially affected by Mn mineral processes like ageing and transformation in sediments[37,45,46,54], and we therefore suggest that Mn mineral processes will also play an important role in their oceanic cycling and have consequences for their mean oceanic concentrations and isotopic compositions. A notable near-sediment source from sediment to seawater is diagnosed for Cu via a data-assimilated model[50] reflecting a benthic flux of Cu, perhaps linked to Mn mineral processes, organic matter remineralisation and lithogenic particle dissolution[55]. Oceanic distributions of such metals are therefore not only affected by internal cycling processes[1–3], for example, where metals are reversibly scavenged by particles in seawater, they are also mediated by Mn mineral processes in sediments. Further consideration of these mineralogical controls will provide a deeper understanding of metal variations within the ocean and through time.

## Methods

### Synthesis of Ni-birnessite samples

Co-precipitated c-disordered Ni-birnessite in which Ni is both surface adsorbed and structurally incorporated was synthesised following the protocol outlined in Atkins et al.[15,33] modified from Villalobos et al.[56]. To produce a Ni-birnessite sample with approximately equal proportions of adsorbed and incorporated Ni, 320 mL of 0.196 M $KMnO_4$ solution was slowly added (5 min total time) to 360 mL of 0.51 M NaOH solution whilst stirring vigorously at room temperature. Then ~3.4 g of $Ni(NO_3)_2$ was added to 320 mL of 0.366 M $MnCl_2$ solution, which was then slowly poured (35 min total time) into the above mixture while stirring vigorously. The suspension was then left to settle for ~4 h. The supernatant was then syphoned and removed. The suspension was then transferred into 250 mL PPCO centrifuge bottles and centrifuged at $3200g$ for 30 min. The slurry was then washed by mixing with 1 M NaCl, centrifuging at $2750g$ for 20 min 5 times. Each time, the supernatant was discarded. The last wash was adjusted to pH 8 and shaken overnight. The slurry was then washed by resuspension in Milli-Q water 10 times, with each suspension centrifuged at $3200g$ for 10 min. The above method was repeated and modified to change the proportion of adsorbed and incorporated Ni. Instead of pouring the $Ni–MnCl_2$ solution for 35 min into the $KMnO_4$ and NaOH mixture, the solution was poured for a total time of 5 min and 3 h. The rate of precipitation affects the amount of trace metal incorporated in minerals[57], and we find that at a slower precipitation rate, there is more Ni incorporated into the mineral and therefore less Ni is adsorbed to the surface. Mineral identity and purity were confirmed by X-ray diffraction using a Bruker D8 Diffractometer with Cu-Kα radiation ($\lambda \approx 0.154$ nm) and a LynxEye detector (Supplementary Information 3). Diffractograms were recorded from 2–90° $2\theta$ with 0.01° $2\theta$ step size and 1085 ms acquisition time. Silicon dioxide was used as an analytical standard. An aliquot of the dried mineral powders was also dissolved with 6 M HCl and analysed for Ni and Mn by inductively coupled plasma mass

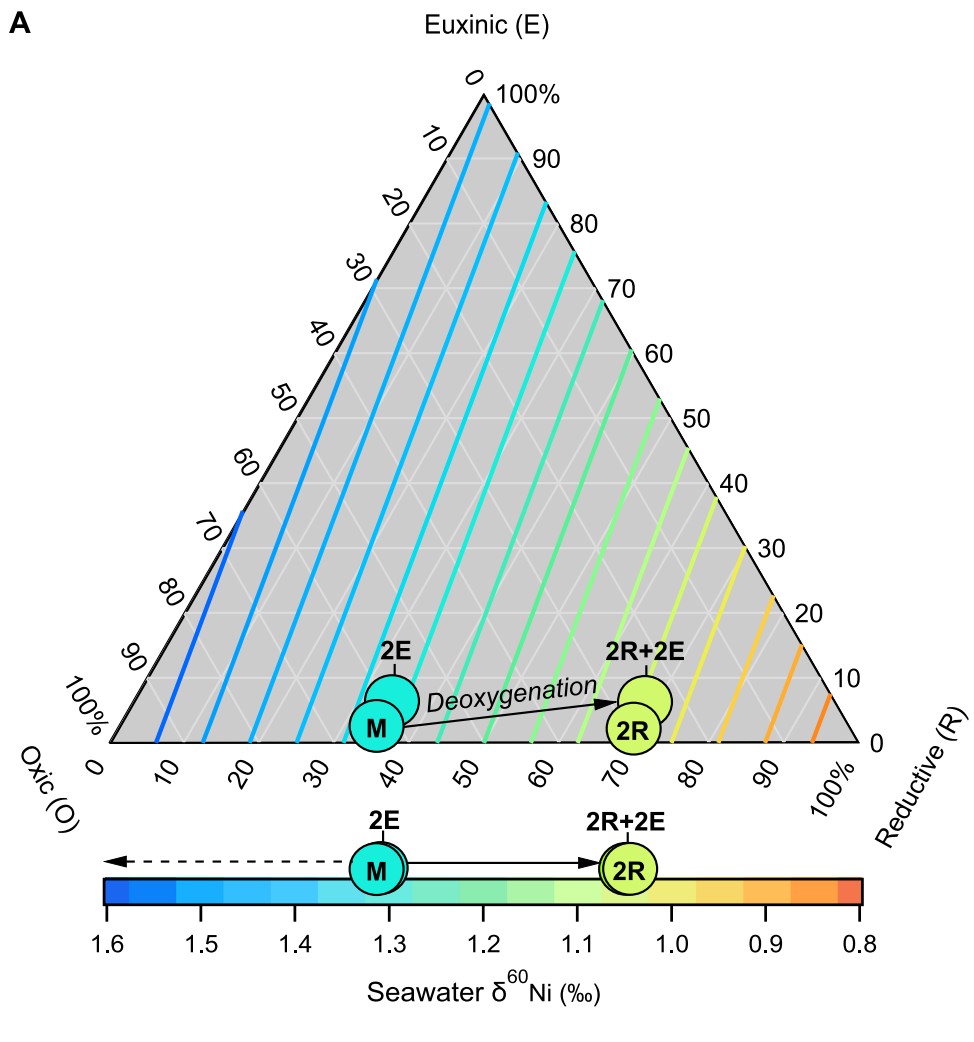

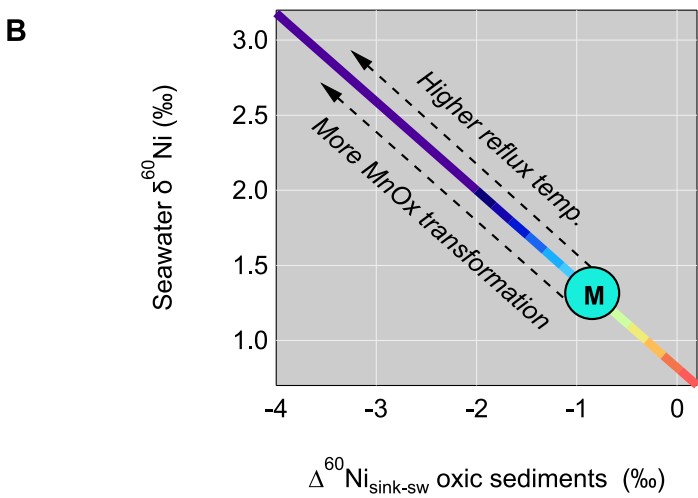

spectrometry (ICP-MS) with a Thermo iCAP Qc ion-coupled plasma mass spectrometer at the University of Leeds. The analytical precision for Mn and Ni was ±1.32% and ±1%, respectively.

**Proportion of adsorbed versus incorporated Ni**
To estimate the proportion of adsorbed versus incorporated Ni in the Ni-birnessite samples and Mn-rich sediments, a wet chemical treatment was performed, in which adsorbed Ni is desorbed into solution, following previously published protocols[36]. Details of the sediment

samples are outlined in Supplementary Information 1. Briefly, 0.01–0.05 g of the Ni-birnessite samples and sediment samples were washed in 200 mL of 0.01 M HCl, and the suspension was shaken at room temperature for 48 h. The suspension was then separated by centrifugation, and the HCl supernatant was collected. The experiment was repeated an additional two times. The solution was then measured for Mn and Ni by ICP-MS with a Thermo iCAP Qc ion-coupled plasma mass spectrometer at the University of Leeds. The analytical precision for Mn and Ni was 1.8% and 0.5%, respectively. Values reported are the

**Fig. 4 | The mass-balance response of seawater Ni isotope composition to mineralogical controls of Ni removal. A** A colour-contoured ternary diagram illustrates the seawater Ni isotope composition as a function of Ni removal by oxic (O), reducing (R), and euxinic (E) sediments (where $fO + fR + fE = 100\%$ Ni sink). Reducing here refers to all settings intermediate between the oxic and euxinic sinks. Euxinic here refers to sediments deposited underlying a water column containing dissolved $H_2S$. End-member isotopic compositions are based on the isotopic offsets between sediment and aqueous phase, detailed in Supplementary Information 4, and M reflects the modern seawater $\delta^{60}Ni$ value. Hypothetical scenarios are used to illustrate seawater $\delta^{60}Ni$ sensitivity to deoxygenation (solid

arrow), where either the proportion of Ni removed by reducing sediments (2R), euxinic sediments (2E), or reducing and euxinic sediments (2R + 2E) is doubled relative to modern. In these scenarios, lighter seawater $\delta^{60}Ni$ values are most sensitive to increases in R. **B** A cross plot shows seawater $\delta^{60}Ni$ values in response to theoretical ranges of Ni isotope fractionation by Mn minerals in oxic sediments. A colour scale for seawater $\delta^{60}Ni$ relationships is used to correspond to values shown in panel A. Values higher than modern seawater $\delta^{60}Ni$ values are theoretically achieved by increased rates of birnessite to todorokite transformation (dashed arrow), which are fundamentally controlled by reflux temperatures at the seafloor.

mean from triplicate experiments, and errors reported are the standard deviation from those experiments. Ni measured from the desorption experiments represents the surface-adsorbed Ni. Incorporated Ni is calculated as the difference between the total Ni and the surface adsorbed Ni. The total Ni of the sample was measured via a total digest of the mineral solid as stated in the section above, or has been measured by other studies[4,13].

The wet chemical desorption treatment was validated using extended X-ray absorption fine structure (EXAFS) spectroscopy conducted at Beamline B18 at the Diamond Light Source Ltd., UK. Three additional Ni-birnessite synthetic samples were prepared as described above, to create a range in the proportion of adsorbed and incorporated Ni. Aliquots of the natural sediment samples USGS Mn nodules NOD A1 and NOD P1 were also selected. The other natural sediment samples contained too low a concentration of Ni for spectroscopic analysis (<0.05 wt%). The three Ni-birnessite synthetic test samples and the two natural sediment aliquots were subject to desorption as described above, and the fraction of adsorbed Ni was compared to that determined by spectroscopy as described below.

Use of the wet chemical treatment to determine the proportion of adsorbed versus incorporated Ni on samples other than those considered in this study (i.e., samples other than Mn-rich sediments and deposits, such as those rich in organic matter or high in carbonates) is not recommend without prior validation.

### Transformation of Ni-birnessite into todorokite

The transformation of Ni-birnessite to todorokite followed the same mild reflux procedure outlined in previous work[15]. In keeping with the previous work, we used a Ni-birnessite sample with similar crystallinity and surface area, and similarly, approximately equal proportions of adsorbed and incorporated Ni (Supplementary Information 3). Crystallinity was compared using X-ray diffraction as described above, surface area was measured as $121 \pm 7\,m^2\,g^{-1}$ (compared to $102 \pm 5\,m^2\,g^{-1}$ in our previous work[15], and adsorbed vs incorporated Ni was measured using wet chemical treatment as described above. Surface area analysis was done with the multipoint Brunauer–Emmett–Teller method using a Micromeritics Gemini VII 2390a Surface Area Analyser. Samples were degassed at room temperature for 24 h using $N_2$ (g) (<1 ppm $CO_2$ (g)) prior to analysis. Graphite was used as an analytical standard. Approximately 35 g of wet c-disordered birnessite slurry was stirred moderately for 18 h in 3 L of 1 M $MgCl_2$ solution. The suspension was centrifuged to a wet paste, then resuspended in 800 mL of 1 M $MgCl_2$ solution in a 1 L round-bottom flask fitted with a glass condenser. The suspension was heated and kept at 100 °C and stirred continuously with a combined heating mantle with magnetic stirrer. Aliquots (~75 mL) were taken at 3 h, 6 h, 24 h, 48 h, 72 h, 1 wk, 2 wk, 3 wk, 4 wk. For each aliquot, the suspension was left to cool and then centrifuged at 3200g for 10 min. The supernatant was filtered through 0.2 μm syringe filters. The solid samples were washed with MQ water and centrifuged. The resulting MQ washes were also filtered for analysis. After 4 weeks, the reflux was stopped, and the final suspension was treated as for the aliquots described above. Nickel lost through washing the mineral accounts for less than 0.1% of the total Ni in the

system, and thus, we conclude that Ni lost through solid washes is negligible to the measured $\delta^{60}Ni$ values of the solid products.

An aliquot of the dried mineral powders from the reflux procedure was also dissolved in 6 M HCl. The supernatant and wash were diluted with 2% HCl. The digested solids were then analysed for Ni and Mn by inductively coupled plasma optical emission spectroscopy (ICP-OES) using a Thermo iCAP 7400 radial ion-coupled plasma optical emission spectrometer at the University of Leeds. The analytical precision for Mn and Ni was ±0.3% and ± 0.9%, respectively. The supernatant and wash were analysed for Ni and Mn by ICP-MS with a Thermo iCAP Qc ion-coupled plasma mass spectrometer at the University of Leeds. The analytical precision for Mn and Ni was ±1.32% and ±1.00%, respectively. Ni and Mn concentrations of the solid, solution and wash are presented in Supplementary Information 5.

### Ni isotope measurements

The mineral solid, solutions and subsequent MQ water washes from each time step of the Ni–birnessite transformation procedure were measured for their Ni isotope ratios. Between 20 and 30 mg of solid samples were digested in 7 M HCl and dried, then re-dissolved in 7 M HCl. Aliquots containing ~200 ng of Ni of the solid and solution samples were then sub-sampled using the concentration data and spiked with ~200 ng $^{61}Ni$–$^{62}Ni$ to obtain a spike to sample-derived Ni ratio of ~1. All samples were then dried and re-dissolved to ensure equilibration of spike and sample Ni.

Nickel was isolated from the sample matrix via a 3-step column chemistry procedure outlined in previous protocols[31,58]. Briefly, anion exchange chromatography was employed for the first two columns. The AG MP-1M anion-exchange resin is used for the first and second columns. The first pass separates Ni from other transition metals in the sample and produces an impure Ni fraction. The second pass through this resin is used to further remove Al and Ti from the Ni fraction. The columns and resins were cleaned with 2% $HNO_3$, then conditioned with 7 M HCl for the first pass and 0.2 M HF for the second pass. The sample, dissolved in 7 M HCl or 0.2 M HF, was then loaded onto the columns, and the corresponding eluates were collected. Before the third column, the sample solution was buffered to pH 5 with ammonium acetate. The final column using the Nobias PA-1 resin separates Ni from Ca, Na and Mg. The third column and resin were cleaned with 1 M HCl, and the matrix elements were eluted using 30 mM $AcNH_4$ and 2 M $NH_4F$, and then Ni was collected with 1 M HCl. Finally, the samples were dried and redissolved in 1 mL of 2% $HNO_3$ for mass spectrometry.

Nickel isotope ratios were measured on a Thermo Scientific Neptune Plus MC-ICP-MS equipped with a Savillex C-Flow PFA nebuliser (50 μl min$^{-1}$) attached to a Teledyne-Cetac Aridus II desolvator at ETH Zürich. The ion masses 56 (Fe), 57 (Fe), 58 (Ni), 60 (Ni), 61 (Ni), 62 (Ni) and 64 (Ni) were measured simultaneously in low-resolution mode with Faraday cups. Mass 56 (Fe) was measured to monitor any interference from 58 Fe on 58 Ni. Instrumental mass bias was corrected using the Ni double spike outlined previously (Cameron et al., 2009; Cameron and Vance, 2014). All Ni isotope compositions are reported

relative to the standard NIST SRM986 in delta notation:

$$\delta^{60}Ni = \left[\frac{(^{60}Ni/^{58}Ni)_{Sample}}{(^{60}Ni/(^{58}Ni)_{NIST\ SRM-986}} - 1\right] \times 1000 \quad (1)$$

Long term reproducibility was monitored by the measurements of USGS standards Nod-A1 and Nod-P1. These give $\delta^{60}Ni = 1.03 \pm 0.05$ ‰ (2 SD, n = 7) for Nod-A1 and $\delta^{60}Ni_{NIST\ SRM986} = +0.35 \pm 0.08$‰ (2 SD, n = 6) for Nod-P1 over the course of this run and parallel studies conducted in the laboratory. Nickel isotope ratios and mass balance calculations for the Ni isotope measurements are presented in Supplementary Information 5.

### EXAFS spectroscopy measurements and data analysis

Spectroscopic data were collected at the Ni K-edge (8.333 keV) on Beamline B18 at the Diamond Light Source Ltd., UK, following previously published protocols[15]. Briefly, samples were presented to the beam as dry powders held between Kapton tape, and data were acquired in fluorescence mode. No photo-redox or visible beam damage was observed on the samples after 5 quick-EXAFS (QEXAFS) scans to $k = 12$ Å$^{-1}$, so 5 QEXAFS spectra were recorded at a single $x,y$ point before moving to a new point to record a further 5 spectra, collecting a total of 30 spectra per sample. Data reduction was performed using the Athena software[59] and for ease of comparison to previously published Ni-birnessite spectra[15,25] were fit in DL-EXCURV[15,25] using a linear combination of hypothetical model clusters representing Ni adsorbed at Mn octahedral vacancies in the phyllomanganate layers and Ni structurally incorporated into the phyllomanganate layers following previously published protocols[15]. Linear combination was performed with a linear combination of the $k^3$-weighted Chi($k$) for each cluster, over the $k$-range 3–12 Å$^{-1}$, where only EF and relative site occupancies were optimised. The errors associated with the optimised site occupancies were evaluated for each fit by assuming that manual changes to the optimised site occupancies were not significant until they generated >10 % increase in the reduced Chi$^2$ function[15,25]. The error on an optimised site occupancy is therefore quoted as the difference between the optimised site occupancy and the site occupancy value incrementally determined to generate a 10 % increase in reduced Chi$^2$.

## Data availability

The experimental data generated in this study are provided in the Supplementary Information, which is also deposited in the general data repository Figshare and can be accessed at https://doi.org/10.6084/m9.figshare.29637206.v1.

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

## Acknowledgements

We thank Cohen Geochemistry Laboratory manager Andrew Hobson and technician Fiona Keay, Aqueous Analysis Facility manager Stephen Reid, and XRD Laboratory manager Lesley Neve (University of Leeds) for laboratory support. We acknowledge the Cohen Geochemistry Laboratory, Aqueous Analysis Facility and XRD Laboratory (University of Leeds), where experimental and analytical work was carried out. We acknowledge Diamond Light Source for EXAFS data collection on beamline B18 (STFC grant no. MG38602-1, C.L.P.), which contributed to the results presented here. We acknowledge the Leeds-York-Hull Natural Environment Research Council (NERC) Doctoral Training Partnership (DTP) Panorama (NE/S007458/1, L.C.). We acknowledge NERC independent research fellowship (NE/P018181/2, S.H.L.). We acknowledge Royal Society Wolfson Research Merit Award (WRM/FT/170005, C.L.P.) and Leverhulme Research Fellowship (RF-2023-395\4, C.L.P). We thank Derek Vance and Sarah Fleischmann for several helpful discussions. We thank the Editor Rebecca Neely and reviewers Bleuenn Guéguen, Brandy Toner and Laura Wasylenki for their insightful comments and suggestions that were extremely helpful to improve our paper.

## Author contributions

L.C. and A.R.D. synthesised the experimental samples and conducted the geochemical experiments. L.C. and C.A. performed the Ni isotope chemistry and isotopic analyses. A.R.D. prepared the EXAFS samples, and C.L.P. conducted the EXAFS data analysis of the samples. L.C. wrote the initial draft of the manuscript with inputs from C.L.P., W.B.H. and S.H.L. All authors reviewed and edited the paper.

## Competing interests

The authors declare no competing interests.
