## [Transparent Peer Review file · Nature Communications]

Mineralogical controls of the oceanic nickel cycle

Corresponding Author: Dr Lena Chen

Version 0:

Reviewer comments:

Reviewer #1

(Remarks to the Author)

The paper presents new nickel isotope data in experimental and natural samples (i.e., respectively synthetic Ni-birnessite and Mn-rich phases in sediments) in order to investigate the effect of mineralogical control on Ni isotope fractionation in Mn-rich minerals such as Mn-oxides.

In their study the authors show the strong effect of Ni bonding mechanisms in Mn minerals on their Ni isotope composition. The model proposed by the authors is that incorporation of adsorbed Ni in Mn minerals in exchange with seawater becomes isotopically heavier with time. Then, post-deposition transformation to todorokite minerals may release isotopically heavy Ni incorporated in the mineral structure. The released Ni can be re-adsorbed with significant isotopic fractionation (i.e., light isotopes would be preferentially adsorbed onto the mineral surface). They observed a relationship between the Ni isotope composition and the fraction of adsorbed Ni and incorporated Ni in the Mn phases. Therefore, the authors explain the range of variability observed in Ni isotope values in Mn minerals by different bonding mechanisms and post-deposition processes during diagenesis.

Nickel isotope composition in various Fe-Mn deposits from modern oceans has already been investigated. In addition, experimental work on Ni isotope fractionation in synthetic Fe- and Mn-oxides has also been performed and published in the literature. All these studies showed the importance of Mn-phases and the oxic sediment sink in the marine biogeochemical cycling of Ni.

Results presented in this study are very interesting in particular because they are a combination of data obtained from experiments and data measured in natural samples. The model presented by the authors is convincing and is supported both by the data acquired on experimental samples and natural samples. The significance of the results to better understanding the biogeochemical cycling of Ni is relevant, more specifically the role of Mn-phases buried in deep sediments on the Ni benthic flux. The paper is very well written and the study well conducted. The data are of good quality and the methods employed suited for the purpose of the study.

I do not have any major comments (just a few minor comments which are detailed below).
This is an excellent study and I am looking forward to seeing it published in Nature Communications.

Detailed comments :

**Figure 1: Typo in the reference year, Sorensen et al., 2020.

In figure 1, one synthetic sample out of three deviates from the mixing relationship, how do you explain this pattern ? Also, this is a small detail but the scaling associated with the y-axis numbers is missing, maybe it would be good to put it back ?

**Line 147-149 : Some Fe-Mn crusts are close to 2 ‰ which is therefore about +0.7 ‰ heavier than seawater. How do you explain these higher values with your model ?

Some explanations are given in the text, for example for isotopically lighter samples (e.g., values obtained at MANOP site M), but not for isotopically heavier samples.

Bleuenn Guéguen, February 2024.

Reviewer #2

(Remarks to the Author)
Please see attached file.

Reviewer #3

(Remarks to the Author)
Synopsis

In this article, the authors combine observations from natural samples with laboratory experiments to propose an interpretation of nickel (Ni) stable isotopes in Earth's oceans. Specifically, the manuscript text addresses the mechanisms by which manganese (Mn) minerals in sediments, crusts, and nodules impart a wide range of stable isotope fractionation outcomes. Explaining greater than 85% of the variance in Ni isotopic fractionation (between aqueous and mineral phases) as a function of a simple acid extraction procedure that estimates adsorbed Ni is indeed a noteworthy result. I think the work is a logical and pleasing extension of the author's past contributions to the literature in the realms of mineral surface chemistry, Ni speciation during adsorption to and incorporation into layer-type Mn oxides (phyllo-manganates), and Ni stable isotopic fractionation in marine systems. Overall, I find the main text, figures, and proposed mechanisms persuasive.

A manuscript like this one is much needed if the Ni stable isotope system is to be of any use in interpreting past oceanographic conditions. The authors point this out directly in the text and I agree with them entirely.

If I understand the situation correctly, if published this manuscript would become the main reference to an accessible and simple wet chemistry method for measuring and interpreting the underlying processes responsible for the Ni stable isotopic signature in the sediment/crust/nodule samples. That would be valuable to the scientific community. It would remove the need for direct spectroscopic observation of the Ni coordination chemistry by synchrotron X-ray absorption spectroscopy. For that potential outcome, I feel that the authors need to strengthen their description and validation of the method. Even if most of that is presented in the supplemental, a validation of the mineralogy and surface chemistry is needed. Also needed is more discussion of the samples for which this method is valid. I expect that the authors have access to everything they need to do this based on their past work.

On the method validation topic, I would like to see guidelines from the authors about when the acid extraction method is valid. Does it work for any marine sediment, or is there some required Mn concentration? Will it work in samples with high organic matter or high carbonates? What are the minimum requirements for demonstrating the mineralogy of the sample to obtain a responsible interpretation? I expect that many geochemists with access to appropriate ICP-MS instrumentation will be eager to apply this new approach – what do they need to know to make responsible measurements and interpretations using this method and proposed mechanistic framework?

Itemized comments

Line 37-38: I don't follow how this statement about reversible 'scavenging' relates to your argument, please explain more directly.

Line 40: Sentence ending on this line needs a reference.

Line 53: Agreed!

Line 93: Not sure that I totally agree with this phrasing. Some of the author's own work over the past 10 years has systematically established the foundation for the work that the authors are now putting forward. The authors should be more specific.

Line 120: Is this method published elsewhere? If this is the first publication and application of the method, more details are needed even if they are mostly in the supplemental.

Line 127-131: Was Ni EXAFS done for any of these synthetic or natural materials? If not, was the new acid rinse method applied to samples for which EXAFS was already in hand? Many geochemists will be excited to apply this method and I think this will be the main citation for the method. More validation of the method and details about how to interpret results is needed.

Line 163: Please add a little more context about the samples (e.g. MANOP site M) in the text so the reader can understand the basics without consulting the supplemental.

Figure 1. The caption should contain more information about the sample names in the legend. I know it makes the caption long, but you want this important figure to be understandable on its own (or nearly so) – it is your main story. For each sample set, how was the fraction of adsorbed and incorporated measured? In the supplemental, this reader also wants more information about the samples. Other than elemental composition, what complementary measurements have been made (Mn, Ni EXAFS; TEM; ...)?

Line 184: "...real..", consider replacing with "natural"

Line 194-196: It is a bit ambiguous whether the details of the production of todorokite is part of this study or relying on past published work. If relying on past published work, how directly comparable were the experimental conditions? Is this a process that is robustly reproducible? I ask because its not clear and, in my experience, we characterize each batch of newly synthesized minerals because the same method won't always produce the same mineral product to the specifications

needed.

Figure 2: The first week or so of the experiment is really dynamic! While panel C suggests simple equilibrium between the mineral and aqueous phase (no changes to [Ni] with time), the fractionation between water and mineral is continuing to drift up at 30 days. It would be helpful to this reader if the authors commented on this in the main text.

Figure 3: When viewing the figure I needed to annotate the $\delta^{60}\text{Ni}$ axis for the top panel (consider giving it a letter A?) with INC/heavier and ADS/lighter. Looking back at it now, I'm not totally sure whether the isotope axis is representing the dissolved for mineral phase. Consider giving the axes for both top and bottom panels more descriptive labels to match the main findings of the manuscript. What is the process or processes leading to remobilization? Reductive dissolution? Or is that the Ni released due to todorokite formation?

Line 253: released to water?

Line 257-260: I'm still confused about this reference. It is enriched and particles but also released from particles?

Line 268-270: Is the seawater isotope composition driving the sediment record? I was thinking it was the minerals in the sediments that were driving the seawater isotope composition.

Line 284: for this broad audience, you might want to define euxinic

Line 304-305: Yes, I agree this is important, thanks for doing this work

Line 345: Standard Bragg diffraction isn't great for identifying phyllosulfates. Were other characterizations done?

Line 535-566: I'd be a lot happier with this new acid extraction method for measuring adsorbed Ni if Ni EXAFS was used to confirm the adsorbed and incorporated fractions for some of the materials presented. Alternatively, some archival samples from past studies could be submitted to the acid extraction and the two methods (EXAFS and acid extraction) compared. It might be that the authors have this type of data from their past work. Presenting that in a revised form to firmly establish the acid extraction method could potentially provide the validation that this author wants to see. If this manuscript becomes the main citation for the method, it should have all of the validation details in the supplemental materials.

Supplemental materials

Supplemental Fig.1 and Fig.2: I think the goal of these XRD measurements is to demonstrate that a Ni-loaded phyllosulfate transforms to a tunnel structure (todorokite) in the reflux experiment. In scrolling between Fig.1 and Fig.2, it is difficult to see this transformation. Very little change in the diffractograms is visible over the 4 week experiment. If we consider that a peak should be ~ 10x the noise level, then these XRD patterns are probably all the same within the noise.

What other solid-state evidence do you have that the reflux experiments promoted transformation from a phyllosulfate to a todorokite? TEM? Mn EXAFS?

Supplementary Fig.2 seems to be missing a panel. The caption refers to "(right)" and "(B)" but those are not in the figure.

Line 41 – Please add the electron microscopy evidence to this supplemental text. Were these events, like changes in crystal size and shape, observed for the samples used in this new manuscript?

Line 51-53 – I agree that the interpretation (phyllosulfate to todorokite) is consistent with the observations reported from a wet chemistry perspective and prior publications by the authors. However, the data currently in the supplemental for the mineralogy (XRD) does not demonstrate the transformation.

Version 1:

Reviewer comments:

Reviewer #2

(Remarks to the Author)

Dear Editors,

Thank you for the opportunity to dig back into this manuscript, which presents some very useful data and innovative ideas about what determines the Ni isotopic composition of Mn-rich marine sediments. Learning how the isotopes work is a crucial step toward figuring out the marine cycle of Ni in the present and past.

The authors have clearly made a thoughtful and dedicated effort to address the weaknesses in the previous manuscript, by conducting EXAFS analyses and making edits to the manuscript that greatly improved clarity and scientific integrity. They have provided more evidence to support their method for separating adsorbed and incorporated Ni and measuring those relative proportions, and they have acknowledged the large uncertainties in the fractions of Ni incorporated.

I truly respect these authors and the work they have put into this paper, and I want to be able to say "accept" now, but I'm afraid a couple of sticky logical problems remain, such that the main conclusion remains poorly supported.

The primary conclusion the authors want to make is that the fraction of structurally incorporated Ni versus adsorbed Ni is the variable that governs isotope fractionation between Ni in solution and Ni on and in Mn oxyhydroxide phases, for samples that have achieved equilibrium.

They did some experiments sorbing Ni to synthetic birnessite in which they intentionally varied the proportions of Ni

adsorbed versus incorporated and looked for a correlation between the solid-solution Ni isotope fractionation and the fraction of Ni adsorbed. The parameter they used to vary the fraction of incorporated versus adsorbed Ni was rate of addition of Ni+MnCl₂ solution to the reactor. They poured the Ni+MnCl₂ solution into the KMnO₄+NaOH solution over either 5 minutes, 35 minutes, or 3 hours, which likely affected precipitation rate and Ni sorption rate and did result in different amounts of adsorbed versus incorporated Ni (which they have now verified with EXAFS). I guess we can assume that the more slowly the Ni was added, the larger the fraction of Ni incorporated, although I am unable to find any information in the main text, figure caption, or supplemental file to confirm that.

In any case, the resulting fractionations from the initial Ni solution define a flat trend in $\delta^{60}\text{Ni}$ versus fraction adsorbed. If the experiments achieved equilibrium, this would be a direct indication that fraction of Ni adsorbed versus incorporated does not govern Ni isotope fractionation. But the experiments almost certainly did not achieve equilibrium, even though the authors describe them as “equilibrium dominated.” The lengths of time the Ni and birnessite were allowed to “see each other” were 5 min+4 hours, 35 min+4 hours, and 3 hours+4 hours. The authors point out that the 24-hour experiments of Sorensen et al. (2021) and the up-to-27-day experiments in Wasylenki et al. (2024), which plot off their main trendline in Figure 1, are off the line because they did not come close to equilibrium and reflect a strong kinetic isotope effect. The authors are also aware from watching our Goldschmidt talks that my own group’s four-month experiments did not achieve equilibrium (they need not cite the abstract, but they are aware of those results). If somehow their 7-hour experiments did reach equilibrium, they would need to provide some evidence of that. It is likely true that co-precipitation of Ni with birnessite should allow greater structural incorporation than Ni sorption to pre-formed birnessite, in the early minutes/hours of the experiment, but it is difficult to imagine in the absence of any evidence that enhanced initial incorporation of Ni would make the time to achieve equilibrium speed up from more than four months to 7-hours.

But the authors go on to combine those experimental points with data from natural samples, which they (fairly safely) assume have achieved equilibrium, and fit a single trendline that they say represents equilibrium fractionation. But those experimental points have no business being included in a trend that supposedly reflects equilibrium fractionation. And the fact that the three experimental points fall near the trendline does not in any way serve as evidence that the experiments achieved equilibrium. (That would be circular reasoning.)

The more likely reasons for the values of $\delta^{60}\text{Ni}$ for those experiments falling close to that trendline are that (1) they were included in the regression and (2) the mechanism by which Ni became structurally incorporated was by first sorbing to the edges of Mn octahedra and then becoming incorporated as the structure was growing, as described by Manceau et al. (2021) for co-precipitation of Ni with birnessite. Because the diffusion step from triple-corner-sharing or edge sites to the incorporated site isn’t necessary if Ni enters the structure as the birnessite is forming, we would not expect to see so much kinetic fractionation arise from incorporation. We might expect to see the smaller fractionation that occurs during triple-corner-sharing and edge-sharing adsorption, which is $\sim -1.5\text{‰}$. And, sure enough, the $\delta^{60}\text{Ni}$ values here are around -1.5‰ , rather than the 3-4‰ observed in other short-duration experiments where diffusion must occur for incorporation of Ni. The authors did not mention or rule out this possible alternative explanation for their experimental results.

Another logical problem pertains to Ni isotopes in ferromanganese crusts. The primary conclusion of the whole paper is that fraction of Ni adsorbed versus incorporated is what governs variability in $\delta^{60}\text{Ni}$ between solution and solid. The anchor author’s pioneering work on Ni in Mn-rich marine sediments demonstrated with EXAFS that three ferromanganese crusts had only structurally incorporated Ni, and, without further information, we might infer that this applies to all crusts, as the authors do in this manuscript. If crusts have only structurally incorporated Ni, however, they should all plot at zero on the x-axis of Figure 1, and they should all have $\delta^{60}\text{Ni}$ near that of seawater, which is $+1.3\text{‰}$. But the ~ 30 crusts analyzed to date have an enormous range of $\delta^{60}\text{Ni}$ values, from $+0.4$ to $+2.4\text{‰}$ (most of the range for all samples formed on Earth). Either crusts do not generally have all their Ni incorporated within phyllosilicate phases, or there is some variable other than fraction of Ni incorporated that has an extremely strong influence on Ni isotope fractionation. The authors chose to refer only to the average of the ~ 30 published analyses, which is, largely by chance (sampling bias), very close to the seawater value. If the authors want to use the crusts to support an argument that fraction of Ni incorporated governs Ni isotopes, they must acknowledge that big range in $\delta^{60}\text{Ni}$ and explain why that does not undermine their central conclusion.

A smaller issue is that the authors tell us that Vance et al. (2016) observed the big, kinetic fractionation ($\sim -4\text{‰}$) between Black Sea MnOx particles and dissolved Ni in the water column because the particles form and dissolve relatively rapidly. But those particles form by co-precipitation and almost certainly have >7 hours to equilibrate. If the experiments in this study represent equilibrium, why would the Black Sea particles not have equilibrated with the water column?

Finally, the paper still does not adequately express big-picture implications of the work in the way that most Nature Comm. papers do. The teaser in the opening paragraph (about how researchers should direct attention to mineralogical controls in their thinking about marine metal cycles in general) is not later matched with thoughts about any big-picture questions that could be answered or theories that might be revised if we all started thinking about mineralogical controls. The Nature Comm. readership is accustomed to seeing some implications that go well beyond the narrow topic of the paper. The specific mineralogical control demonstrated in this paper is that birnessite and todorokite have different proportions of surface sites and incorporated sites. This is certainly important for figuring out Ni cycling, and it assists ideation about a Ni paleoredox proxy, but are there any other examples where mineralogical controls might be important? (Sorption mechanisms are unusually complex for Ni+birnessite.) Are there any misconceptions or controversies or sticking points in the literature that could be resolved by consideration of mineralogical controls? The authors are encouraged to make the paper more appropriate for a high-profile publication by strengthening it in this regard.

Best regards,

--Laura Wasylenki

Attachment: Word file containing specific comments on the manuscript.

Reviewer #3

(Remarks to the Author)

Hello! I have reviewed the authors' revisions. I think they responded to my comments, and the comments from the other reviewers, thoroughly and clearly. I have no further requests for changes to the manuscript and congratulate them on an important contribution. Brandy Toner

Version 2:

Reviewer comments:

Reviewer #2

(Remarks to the Author)

Dear Rebecca,

I have read carefully the rebuttal letter submitted by Chen et al., and I am now satisfied that the authors have produced a manuscript that is ready for publication. I especially appreciate the clear text presented in Lines 225-269 and the thoughts behind it. Possible complications with the interpretation are now properly acknowledged, and the Manceau work is cited and discussed. The addition of the paragraph at the end about other trace metals is another welcome improvement.

I did not read the entire manuscript closely, but I found a few tiny mistakes as I re-read critical sections that can be easily corrected.

Line 178: "... these data...." Same in Lines 185 and 195 and throughout.

Line 191: Change semicolon to comma.

Line 233: Insert comma after "Furthermore."

Thanks again for the opportunity to think hard and learn from this manuscript on my favorite topic!

Regards,

—Laura
